# Determinants and consequences of heavy episodic drinking among female sex workers in Ethiopia: A respondent-driven sampling study

Minilik Demissie Amogne[1,2]*, Anette Agardh[1], Ebba Abate[2], Jelaludin Ahmed[3], Benedict Oppong Asamoah[1]

**1** Department of Clinical Sciences Malmo, Social Medicine and Global Health, Lund University, Malmo, Sweden, **2** Ethiopian Public Health Institute, Addis Ababa, Ethiopia, **3** CDC Ethiopia, Addis Ababa, Ethiopia

* minilik_demissie.amogne@med.lu.se, minewdem@gmail.com

## Abstract

### Background

Female sex workers (FSW), due to their working conditions, have an increased likelihood of heavy episodic drinking (HED), which is associated with risky sexual behavior. Nevertheless the specific contribution of HED to risky sexual behavior among FSWs in Ethiopia is not well documented for prevention activities.

### Objective

The purpose of this study was to explore the determinants and consequences of HED among FSWs in Ethiopia.

### Methods

A cross-sectional study using respondent-driven sampling was conducted among 4886 FSWs in 11 major towns in Ethiopia in 2014. A structured interview was performed, and data were examined using descriptive statistics and multiple logistic regression analyses.

### Results

Most (66%) FSWs consumed alcohol, and the prevalence of HED was 29.1%. Compared to street-based FSWs, those who worked in bars/hotels and local drinking houses had 2.19 and 1.29 times higher odds of HED, respectively. FSWs who started selling sex when younger than 18 years (compared to those who started when older than 25 years) and those who were forced into selling sex had 1.48 and 2.91 times higher odds of HED, respectively. FSWs with more income from selling sex and FSWs who chewed khat reported increased odds of HED. Moreover, FSWs with experience of HED reported 1.27 and 1.44 times higher odds of physical beating and condom breakage/slippage, respectively. Furthermore, the population attributable risk fraction of HED among FSWs showed that 6.2% of physical beating and 8.9% of condom breakage/slippage could be attributed to HED.

**Data Availability Statement:** The data is owned by Ethiopian public health institute and is available based on request. Anyone who wants to get data

from the institute should write email to the director general (Dr. Ebba) copying the director of data management (Dr. Alemnesh). The email should be brief on the objective of the study and the data set to obtain; future information will be followed after the first contact. Their details are below. Director General: Dr. Ebba: ebbaabate@yahoo.com Dr. Alemnesh: alemmirkuzie@yahoo.com.

**Funding:** The study was funded by the President's Plan for AIDS Relief (PEPFAR) through Ethiopian Public Health Association (EPHA) under the terms of PS001229.

**Competing interests:** The authors have declared that no competing interests exist.

## Conclusion

In general, several factors increase the experience of HED, and HED in turn increases the likelihood of violence and condom breakage. These factors could inform programs and intervention activities among FSWs populations.

## Introduction

Excessive use of alcohol has serious social, economic, and health-related consequences [1, 2]. When compared with men, women are more vulnerable to the adverse consequences of alcohol use such as violence, unintended pregnancy etc. [3]. Women who are engaged in a stigmatized profession such as selling sex may be at increased risk of the negative health effects of alcohol use [4].

The harms associated with alcohol depend on the volume, frequency of consumption, and contextual factors [5, 6]. Most FSWs work in alcohol-serving venues (hotels, bars and local drinking houses), on the street, or in red light houses. Because of their working situation, environment, and other factors, FSWs are more exposed to heavy episodic drinking (HED); for example, FSWs who work in alcohol-serving venues are expected to facilitate alcohol sales [6–11].

Previous studies show that younger FSWs experience more HED than older FSWs. A study in China reported that FSWs who started selling sex at a younger age experienced more HED [11]. FSWs forced into the sex trade also may have increased vulnerability to abusing alcohol and other substances as a coping mechanism [12, 13].According to studies in Kenya, Philippines and the US Virgin Islands, FSWs who reported alcohol abuse were more likely to report having used other drugs [10, 14, 15].

In Ethiopia, the stimulant leaf khat (Catha edulis) is popular, although chronic use of khat or high doses may have adverse health and socioeconomic consequences [16]. Khat chewing may contribute to reduced productivity, loss of working hours, malnutrition, and diversion of income, and for chronic users it is associated with hypertension, insomnia, liver toxicity, oral cancer, loss of appetite, and gastrointestinal effects [17]. When a stimulant such as khat is combined with alcohol, it might mask the effect of alcohol, thereby increasing the chance of HED [18]. Young people often chew khat during the daytime and go to bars to drink alcohol at night to purposely reduce the stimulant effect [19]. In Ethiopia, FSWs often chew khat to pass the time during the day and to increase energy during the night for sex work and to socialize with other FSWs [19].

The consequences of HED for FSWs are not merely limited to the physical effects of intoxication on the body but also increase the risk of violence and risky sexual behaviors [15, 20–22]. Studies among FSWs from Malawi, Kenya, China, and South India reported an association between alcohol use and inconsistent condom use [15, 23–26]. Another study among FSWs in Kenya also reported that HED (33%) in the past month was significantly associated with unprotected sex, violence, and sexually transmitted infections [6]. Yet, in the aforementioned study and another study in Kenya, HED (having ≥5 drinks) was not found to be significantly associated with HIV infection [6, 10].

Moreover, excessive alcohol use can impair FSWs' abilities to detect the risk of violence and increases their involvement in risk-prone situations [27, 28]. A study in China reported that FSWs who drank alcohol before sex were significantly more at risk for sexual coercion from their clients [29]. In addition, a study in Kenya and South Africa also showed that reducing alcohol consumption among FSWs significantly decreased violence [30, 31].

In general, studies among FSWs conducted in different countries found that the venue, income, FSW age, and other related factors were associated with HED among FSW, and those with HED in turn were exposed to violence and risky sexual behaviors [6, 20, 24, 31].

However, little is known about the determinants and consequences of HED among FSWs in Ethiopia, although few studies have investigated the potential role of socio-demographic and other contextual factors [32, 33]. In addition, the relationship between HED and violence and risky sexual behaviors is not well documented in Ethiopia. Therefore, the purpose of this cross-sectional study was to explore the determinants and consequences of HED among FSWs in Ethiopia.

## Methods

### Study design

This analysis was part of a larger cross-sectional study concerning HIV prevalence and related risk factors among FSWs in Ethiopia in 2014. Respondent driven sampling (RDS) was used to collect the data. RDS is a chain referral sampling technique recommended for hard-to-reach populations with a mathematical model that weights the sample [34].

### Study area, period, and population

The study was conducted in seven major regional towns (Addis Ababa, Mekelle, Bahir Dar, Adama, DireDawa, Gambela, and Hawassa) and four main transport corridor towns in Ethiopia (Metema, Kombolcha, Semera-Logia, and Shashemene). This analysis included all data collected from FSWs living in these 11towns.

For the purposes of this study, we defined FSW as women who practice sexual activity with the preconditions of financial or in-kind benefits. FSWs were included in the study if they received money or other benefits for sex with ≥4 people within the last 30 days, were aged≥15 years, were properly recruited by a peer (presented the coupon), and gave consent both for the interview and blood sample collection.

### Sample size

A minimum sample size of 400 was calculated for each town using anticipated HIV prevalence of 25%, 6%precision, 95%confidence interval (CI), and design effect of two. Because of the RDS sampling requirement for equilibrium, the number of FSWs who participated in each town was not exactly 400 and the total number of FSWs who participated was 4886.

### Data collection procedure

Six FSWs, called seeds, were selected to initiate coupon-based recruitment in each town. Seeds were selected purposively to represent the geographical and occupational (e.g., brothel vs. street-based) diversity of the target populations. Seeds were identified through formative assessments (key informant interviews and in-depth interviews) with key stakeholders and representatives of different key population groups.

Following informed consent, participants were interviewed in a private room by a nurse counselor with a structured questionnaire in the Amharic language and were asked to provide blood samples for HIV testing. The questionnaire was piloted before the actual implementation of the study in town that was not included as a study site. Subsequently, participants were provided with up to three coupons and instructed to recruit their FSW peers into the study. FSWs were given a primary incentive of 100 ETB (US$5.00) and an additional 50 ETB (US $2.50) for each eligible peer recruited and enrolled into the study. The exchange rate of ETB to USD is according to 2014 exchange rate. Fingerprint recognition software was used to create a

unique study identification number for each participant to help prevent the same participant from enrolling more than once.

## Variables

According to World Health Organization criteria, HED is defined as "the proportion of adults (15+ years) who have had at least 60 grams or more of pure alcohol on at least one occasion in the past 30 days. A consumption of 60 grams of pure alcohol corresponds approximately to 6 standard alcoholic drinks [35]. Accordingly, the survey tool asked the following question: "How often do you have 6 or more drinks on one occasion?" A drink was defined as a drink of beer, liquor, or other local drinks such as tella, tej, or areke. The response alternatives were: 0, Never; 1, Less than once a month; 2,Monthly; 3, Weekly; and 4, Daily or almost daily. For the purpose of this analysis, respondents who selected 2 through 4 were categorized as engaging in monthly HED. Those who never drank were grouped with those who did not have HED.

To examine potential determinants of HED, we used the following independent variables: current age, age when the participant started selling sex, whether participant was forced to sell sex, whether participant supported others (financial or other support), monthly income from selling sex, educational status, sex selling venue, khat chewing frequency, and ever used any other drugs. Both the independent and dependent variables were selected based on previous evidence and on the objectives of the study.

To examine the potential consequences of HED, we included the following measures as dependent variables: HIV status, violence, condom breakage/slippage, and inconsistent condom use. During multiple regression analyses, each dependent variable was adjusted for age (sex selling starting age), average income, sex selling venue, khat chewing, being forced to sell sex, and educational level. Current age was a continuous variable and was categorized in 10-year intervals: younger age (15–24 years), middle (25–34 years), and older (≥35 years), with the younger age category used as a reference group. Age at start of sex work was also a continuous variable and was categorized as minor (<18 years), younger (18–24 years), and older (≥25 years). Monthly income from selling sex was an open-ended question and was categorized as <1000 ETB (<US$50), 1001–2000 ETB (US$51–$100), 2001–3000 ETB (US$101–$150), 3001–4000 ETB (US$151–$200), 4001–5000 ETB (US$201–$250), and >5000 ETB (>US$251), based on the cost of living in Ethiopia. Educational status was categorized as no formal education, primary first cycle (grade 1–4), primary second cycle (grade 5–8), and secondary and above, based on Ethiopia's education system. FSWs were categorized according to the primary location of work, including street-based, red light houses, and bar/hotel. Khat chewing frequency (days per week)was also assessed: never, less than once, 1–2 days/week, 3–4 days/week, and 5–7 days/week. Information was obtained about ever using any other drug besides alcohol and khat during the last 30 days and was categorized as "yes" or "no." Using any other drug at least once was categorized as "yes."

Violence was defined as physical beating during the past year. Responses were dichotomized as "yes" or "no" for analysis. FSWs who reported a physical beating at least once were considered exposed to violence. Condom breakage/slippage and inconsistent condom use were assessed in terms of the last 30 days before the study. Inconsistent condom use was measured with the question, "With how many different paying partners did you have sex without condoms in the last 30 days?" Not using condoms at least once was considered inconsistent condom use.

## Data analysis

Statistical analysis was performed using SPSS, version 20 (Chicago, IL). Descriptive statistics were used to provide summary measures (means and frequencies). For each record in the data

set, the RDS-based weights were applied. RDS weights were generated using RDS-Analyst software. In addition to this weight, information on estimated number of FSWs in each town, as well as number of completed interviews in each town was applied to obtain a final weight.

Crude and adjusted odds ratios with 95% CI were obtained using bivariate and multivariate logistic regression analyses models. Before conducting multivariate analysis, we performed correlation analysis to examine potential multicollinearity. P-values <0.05 were considered statistically significant, and cases with missing data were excluded from the analyses.

## Estimation of HED-attributable risk fraction among HED exposed and total FSW population

We estimated attributable risk fractions (AF) of HED for the occurrence of physical beating, condom breakage/slippage, and inconsistent condom use among FSWs (who reported HED) and the entire FSW population. The proportions were calculated to measure the effects that could be avoided if HED was prevented under an assumption of a causal link between HED and the listed variables.

Attributable risk fraction (AF) among HED-exposed FSWs was calculated as: [36]

$$AF = 100(OR - 1)/OR$$

The population AF (PAF) among total FSW population was calculated as:

$$PAF = P \cdot AF = 100P \cdot (OR - 1)/OR$$

where OR is the odds ratio generated from multivariate logistic regression analysis and P is the proportion of HED among the FSW population.

## Ethical considerations

The data set was obtained from the Ethiopian Public Health Institute (EPHI). The protocol was cleared at the Scientific and Ethical Research Office (SERO) of EPHI, Ethiopian Science and Technology Ministry Ethical Committee, and CDC-IRB. Individual written informed consent was obtained from each participant for the interview and blood sample collection before the study was conducted. Permission was obtained from the ethical review committee to collect consent from FSWs between the ages of 15 to 18 because they are considered to be emancipated minors. For FSW under the age of 18 who indicated need for additional service, procedures were in place to refer to relevant service providers.

## Results

### Socio-demographic characteristics

A total of 4886 FSWs participated in the study, and most (59.0%) were aged 15–24 years, with a mean age of 24.17years (standard deviation [SD], 5.94). Most (59.1%) FSWs started selling sex between the ages of 18 and 24 years and 55.9% attended primary first or second cycle education. Of the participants, 38.3% sold sex on the street, 59.9% were never married, and 32.8% earned a monthly average income of less than 1000 ETB (US$50; Table 1).

### Behavioral and other related factors

Of the participants, 30.4% reported that they were forced into the sex trade and48.9% supported others (financial or other support). Of the participants, 26.8% reported condom breakage/slippage within the past 30 days before the study, and 20.7% were HIV positive. Of the participants, 19.6% reported physical beating within the past 12 months. 52.9% chewed khat at

**Table 1. Distribution of socio-demographic and other background characteristics among female sex workers in 11 towns in Ethiopia (2014).**

| Variable | Frequency | Percentage |
|---|---|---|
| **Age**, years | | |
| 15–24 | 2882 | 59.0 |
| 25–34 | 1588 | 32.5 |
| ≥35 | 416 | 8.5 |
| Total | 4886 | 100.0 |
| Missing | 0 | |
| **Mean age = 24.17 years SD = 5.94** | | |
| **Educational status** | | |
| No Education | 1228 | 25.2 |
| Primary 1st cycle (grade 1–4) | 780 | 16.0 |
| Primary 2nd cycle (grade 5–8) | 1942 | 39.9 |
| Secondary and above | 923 | 18.9 |
| Total | 4873 | 100.0 |
| Missing | 13 | |
| **Sex-selling venues** | | |
| Street | 1874 | 38.3 |
| Local drinking houses | 868 | 17.8 |
| Spa/Massage/Beauty salon/Own house | 159 | 3.2 |
| Red light houses | 476 | 9.7 |
| Bar/Hotel | 1143 | 23.4 |
| Others | 367 | 7.5 |
| Total | 4886 | 100.0 |
| Missing | 0 | |
| **Current marital status** | | |
| Never Married | 2925 | 59.9 |
| Married/Cohabited | 57 | 1.2 |
| Separated/Divorced | 1753 | 36.0 |
| Widowed | 143 | 2.9 |
| Total | 4878 | 100.0 |
| Missing | 8 | |
| **Age when started to sell sex, years** | | |
| <18 | 1216 | 24.9 |
| 18–24 | 2889 | 59.1 |
| ≥25 | 781 | 16.0 |
| Total | 4886 | 100.0 |
| Missing | 0 | |
| **Mean age when started to sell sex = 20.66** | | |
| **SD = 5.31** | | |
| **Average monthly income from selling sex** | | |
| ≤1000 ETB (≤$50) | 1603 | 32.8 |
| 1001–2000 ETB (US$51–$100) | 1574 | 32.2 |
| 2001–3000 ETB (US$101 –$150) | 929 | 19.0 |
| 3001–4000 ETB (US$151–$200) | 491 | 10.1 |
| 4001–5000 ETB (US$201 –$250) | 167 | 3.4 |
| >5000 ETB (>US$251) | 122 | 2.5 |
| Total | 4886 | 100.0 |
| Missing | 0 | |

SD-Standard deviation

least once per week, and 10.8% used other drugs. Furthermore, 66% of FSWs drank alcohol, and the prevalence of HED among all FSWs was 29.1% (Table 2).

### Determinants of HED in bivariate logistic regression analysis

Bivariate logistic regression was used to identify significant determinants of HED. Variables significantly associated with monthly HED experience included educational status, monthly income, sex selling venue, age when started selling sex, being forced into selling sex, providing regular support to others (financial or other), and frequency of khat chewing (Table 3).

### Determinants of HED on multivariate logistic regression analysis

In multivariate logistic regression analysis, having a monthly income greater than 1000 ETB (US$50) was a significant predictor of HED; as income increased, the odds of HED also increased. FSWs who worked in bars/hotels (adjusted odds ratio [aOR], 2.19[95% CI: 1.81–2.66]) and in local drinking houses (aOR, 1.29 [95% CI: 1.002–1.67]) had higher odds of HED compared to those who worked on the street. FSWs who worked in spas, massage parlors, beauty salons, or their own house (aOR, 0.46 [95% CI: 0.25–0.83]) had lower odds of HED than street-based FSWs. FSWs who started selling sex before the age of 18 years had 1.48 (1.13, 1.95) and between 18–24 years 1.59 (1.25, 2.03) times higher odds of HED than FSWs who started selling sex after the age of 25 years. FSWs who were forced into the sex trade had 2.91 (2.45, 3.46) times increased odds of HED than FSWs who chose to start selling sex. In addition, FSWs who reported chewing khat at least once per week had higher odds of HED than non-users; as the number of days of chewing khat increased, the odds of HED also increased. Educational status and those who reported support for others were no longer significant when adjusted for other variables (Table 4).

### Consequences of HED in bivariate and multivariate logistic regression analysis

Bivariate and multivariate logistic regression analyses were performed using HED as a predictor and condom breakage/slippage, inconsistent condom use, physical beating, and HIV status as dependent variables. In bivariate analysis, HED was significantly associated with each dependent variable except for HIV status. In multivariate analysis, FSWs with HED had 1.27times higher odds of physical beating than those without HED. FSWs with HED had 1.44 times higher odds of condom breakage/slippage than those without HED (Table 5).

### AF of HED on outcome variables

Table 6 shows the HED AF among the FSWs with HED and the entire FSW population. The proportion of HED in the total FSW population was 29.1% and was used to calculate the PAF. The results showed that 6.2% of the risk of physical beating and 8.9% of condom breakage/slippage could be attributed to HED among FSWs.

## Discussion

To our knowledge, this is the first study of FSWs that used RDS and includes 11 towns in Ethiopia. Alcohol use was common among FSWs, and the prevalence of HED was high (29.1%). Factors such as starting to sell sex before the age of 25years, being forced into selling sex, working in a bar/hotel, having a higher income, and chewing khat frequently were significant determinants of HED. In turn, HED was significantly associated with physical beating and condom breakage.

**Table 2. Behavioral and other related factors among female sex workers in 11 towns in Ethiopia (2014).**

| Variable | Frequency | Percentage |
|---|---|---|
| **Someone forced you into selling sex** | | |
| No | 3401 | 69.6 |
| Yes | 1485 | 30.4 |
| Total | 4886 | 100.0 |
| Missing | 0 | |
| **Provide any regular financial or other support to family member and/or to others** | | |
| No | 2496 | 51.1 |
| Yes | 2390 | 48.9 |
| Total | 4886 | 100.0 |
| Missing | 16 | |
| **Inconsistent condom use in the past 30 days** | | |
| No | 4521 | 92.5 |
| Yes | 365 | 7.5 |
| Total | 4886 | 100.0 |
| Missing | 0 | |
| **Condom breakage/slippage in the past 30 days** | | |
| No | 3572 | 73.2 |
| Yes | 1307 | 26.8 |
| Total | 4879 | 100.0 |
| Missing | 7 | |
| **Frequency of alcohol consumption** | | |
| Never | 1660 | 34.0 |
| Once per month or less | 208 | 4.3 |
| 2–4 times per month | 494 | 10.1 |
| 2–3 days per week | 1384 | 28.3 |
| 4 or more days per week | 1140 | 23.3 |
| Total | 4886 | 100.0 |
| Missing | 0 | |
| **Frequency of khat chewing per week** | | |
| Never | 2302 | 47.1 |
| Less than once | 545 | 11.2 |
| 1–2 days | 391 | 8.0 |
| 3–4 days | 283 | 5.8 |
| 5–7 days | 1364 | 27.9 |
| Total | 4885 | 100.0 |
| Missing | 0 | |
| **Ever used any other drugs** | | |
| No | 4358 | 89.2 |
| Yes | 528 | 10.8 |
| Total | 4886 | 100.0 |
| Missing | 0 | |
| **Monthly heavy episodic drinking (HED)** | | |
| No | 3466 | 70.9 |
| Yes | 1420 | 29.1 |
| Total | 4886 | 100.0 |
| Missing | 0 | |
| **Physically beaten in the past 12 months** | | |

(*Continued*)

**Table 2.** (Continued)

| Variable | Frequency | Percentage |
|---|---|---|
| No | 3930 | 80.4 |
| Yes | 955 | 19.6 |
| Total | 4885 | 100.0 |
| Missing | 1 | |
| **HIV status** | | |
| Negative | 3857 | 79.3 |
| Positive | 1009 | 20.7 |
| Total | 4866 | 100.0 |
| Missing | 20 | |

A national household-based study in Ethiopia reported a 1-month HED prevalence of 12.4% (men, 20.5%; women, 2.7%) [37]. Although these findings are not directly comparable with our results, the difference in monthly HED prevalence observed among the general female population and FSWs in our study is considerable and supports the disproportional risk of HED among FSWs. The HED prevalence in our study (29%) was comparable with that among FSWs in Kenya (33%) [6].

HED was significantly associated with average monthly income in our study, which supports findings from a study conducted in Malawi, showing that alcohol use facilitates FSWs' solicitation with clients and is a help in price negotiation [23].

Furthermore, the type of venue used by FSWs to sell sex played a role in alcohol drinking and increased HED. Our study showed that FSWs who worked in a bar/hotel or local drink house had higher odds of HED than street-based FSWs. FSWs working in these venues are expected to facilitate the alcohol sales, which in turn increases HED. Similar studies conducted in Kenya, India, and Indonesia and a meta-analysis involving several countries reported that higher alcohol consumption is more prevalent in a setting where alcoholic drinks are sold [6–10].

Early adolescent alcohol use is associated with an earlier age of first sexual intercourse and greater number of sexual partners [38]. Our study showed that FSWs who started selling sex before the age of 18 years experienced more HED. A similar study conducted in China also reported that FSWs who had their sexual debut at a younger age were significantly more often intoxicated with alcohol before having sex with their client than older FSWs [11]. Although we were unable to determine whether FSWs started drinking alcohol before or after they started selling sex, starting to sell sex at a younger age increased the odds of HED. This association underlines the importance of alcohol risk reduction interventions targeting young FSWs.

Moreover, being forced into sex work increases the vulnerability(risk for rape, inconsistent condom use, etc.) of women/girls by removing their basic autonomy [13]. These FSWs tend to use substances including alcohol as a coping mechanism, which subsequently increases their risk to become a chronic user [12, 13]. Our study also shows that FSWs who reported being forced into selling sex were significantly more likely to experience HED. To address the problem of sex trafficking and its consequences, it is crucial to identify the individuals behind such trafficking and to involve law enforcement officers in these efforts. Another strategy could be to increase and improve the knowledge and ability of law enforcement professionals to identify victims of sex trafficking.

A study conducted among frequent khat users in Addis Ababa, the capital of Ethiopia, reported that khat use was a significant predictor of harmful drinking [19]. Although that study is not directly comparable with our study, our results showed that FSWs who chewed

**Table 3. Bivariate logistic regression analysis results of monthly Heavy Episodic Drinking (HED) in relation to predictor variables among female sex workers in 11 towns in Ethiopia.**

| Variables | N | OR (95%CI) | P-Value |
|---|---|---|---|
| **Age, years** | | | |
| 15–24 (ref) | 2880 | 1.0 | |
| 25–34 | 1590 | 1.09 (0.95, 1.25) | 0.205 |
| ≥35 | 416 | 0.85 (0.67, 1.09) | 0.182 |
| **Educational level** | | | |
| No Education (ref) | 1230 | 1.0 | |
| Primary 1st cycle (1–4) | 780 | 1.23 (0.98, 1.53) | 0.069 |
| Primary 2nd cycle (5–8) | 1941 | 1.97 (1.66, 2.34) | < 0.001 |
| Secondary and above | 922 | 3.05 (2.51, 3.70) | < 0.001 |
| **Average monthly income from selling sex** | | | |
| ≤1000 ETB (≤US$50) (ref) | 1608 | 1.0 | |
| 1001–2000 ETB (US$51–$100) | 1575 | 1.90 (1.60, 2.25) | < 0.001 |
| 2001–3000 ETB (US$101–$150) | 927 | 3.66 (3.04, 4.40) | < 0.001 |
| 3001–4000 ETB (US$151–$200) | 489 | 3.14 (2.51, 3.93) | < 0.001 |
| 4001–5000 ETB (US$201–$250) | 166 | 6.00 (4.30, 8.36) | < 0.001 |
| >5000 ETB (>US$251) | 122 | 4.07 (2.79, 5.95) | < 0.001 |
| **Sex-selling venues** | | | |
| Street(ref) | 1866 | 1.0 | |
| Local drinking houses | 872 | 0.45 (0.36, 0.55) | < 0.001 |
| Spa/Massage/Beauty salon/Own house | 159 | 0.26 (0.15, 0.45) | < 0.001 |
| Red Light houses | 477 | 0.55 (0.42, 0.70) | < 0.001 |
| Bar/Hotel | 1145 | 1.70 (1.45, 1.98) | < 0.001 |
| Others | 367 | 1.80 (1.43, 2.26) | < 0.001 |
| **Age when started to sell sex, years** | | | |
| <18 | 1216 | 1.69 (1.37, 2.10) | < 0.001 |
| 18–24 | 2889 | 1.87 (1.54, 2.26) | < 0.001 |
| ≥25 (ref) | 781 | 1.0 | |
| **Someone forced you into selling sex** | | | |
| No (ref) | 3404 | 1.0 | |
| Yes | 1482 | 5.04 (4.41, 5.76) | < 0.001 |
| **Provide any regular financial or other support** | | | |
| No | 2494 | 2.05 (1.81, 2.33) | < 0.001 |
| Yes (ref) | 2392 | 1.0 | |
| **Frequency of khat chewing per week** | | | |
| Never(ref) | 2304 | 1.0 | |
| Less than once | 546 | 2.03 (1.58, 2.61) | < 0.001 |
| 1–2 days | 390 | 3.67 (2.84, 4.73) | < 0.001 |
| 3–4 days | 282 | 4.59 (3.47, 6.06) | < 0.001 |
| 5–7 days | 1364 | 14.84(12.48, 17.65) | < 0.001 |

Abbreviations: OR, crude odds ratios; CI, confidence intervals.

khat more frequently also experienced more HED. According to different studies, when a stimulant, such as khat, is combined with alcohol, it masks alcohol's effects, so that people cannot assess their level of intoxication, which can result in over-consumption [18, 39]. Therefore, targeting khat chewing among FSWs is an essential part of strategies to reduce HED among FSWs.

**Table 4. Multivariate logistic regression analysis of monthly Heavy Episodic Drinking (HED) in relation to predictor variables among female sex workers in 11 towns in Ethiopia (2014).**

| Variables | N | aOR (95%CI) | P-value |
|---|---|---|---|
| **Educational level** | | | |
| No education(ref) | 1230 | 1.0 | |
| Primary 1st cycle (1–4) | 780 | 1.12 (0.86, 1.46) | 0.407 |
| Primary 2nd cycle (5–8) | 1941 | 1.46 (1.18, 1.80) | < 0.001 |
| Secondary and above | 922 | 2.20 (1.72, 2.81) | < 0.001 |
| **Average monthly income from selling sex** | | | |
| ≤1000 ETB (≤US$50)(ref) | 1608 | 1.0 | |
| 1001–2000 ETB (US$51–$100) | 1575 | 0.99 (0.81, 1.23) | 0.961 |
| 2001–3000 ETB (US$101–$150) | 927 | 1.49 (1.18, 1.88) | 0.001 |
| 3001–4000 ETB (US$151–$200) | 489 | 1.36 (1.02, 1.80) | 0.034 |
| 4001–5000 ETB (US$201–$250) | 166 | 3.94 (2.64, 5.89) | 0.001 |
| >5000 ETB (>US$251) | 122 | 1.46 (0.92, 2.32) | 0.112 |
| **Sex-selling venues** | | | |
| Street(ref) | 1866 | 1.0 | |
| Local drinking houses | 872 | 1.29 (1.002, 1.67) | 0.051 |
| Spa/Massage/Beauty salon/Own house | 159 | 0.46 (0.25, 0.83) | 0.010 |
| Red light houses | 477 | 0.81 (0.60, 1.10) | 0.181 |
| Bar/Hotel | 1145 | 2.19 (1.81, 2.66) | 0.001 |
| Other | 367 | 1.62 (1.22, 2.15) | 0.001 |
| **Age when started to sell sex, years** | | | |
| <18 | 1216 | 1.48 (1.13, 1.95) | 0.005 |
| 18–24 | 2889 | 1.59 (1.25, 2.03) | < 0.001 |
| ≥25 (ref) | 781 | 1.0 | |
| **Someone forced you into selling sex** | | | |
| No(ref) | 3404 | 1.0 | |
| Yes | 1482 | 2.91(2.45, 3.46) | < 0.001 |
| **Provide any regular financial or other support to family member and/or to others** | | | |
| No | 2494 | 1.17 (0.99, 1.38) | 0.070 |
| Yes (ref) | 2392 | 1.0 | |
| **Frequency of khat chewing per week** | | | |
| Never(ref) | 2304 | 1.0 | |
| Less than once | 546 | 1.99 (1.52, 2.59) | < 0.001 |
| 1–2 days | 390 | 3.31 (2.51, 4.36) | < 0.001 |
| 3–4 days | 282 | 4.62 (3.42, 6.24) | < 0.001 |
| 5–7 days | 1364 | 11.15 (9.20, 13.50) | < 0.001 |

Abbreviations: aOR, adjusted odds ratios; CI, confidence interval.

Even though condom use was high in our study, reported condom breakage/slipping was also high. FSWs who reported condom breakage/slipping and inconsistent condom use reported significantly more HED, and this is a major concern for HIV/sexually transmitted infection prevention programs. Our findings are similar to those of studies conducted in India and South Africa [40, 41]. Although an individual may know how to use a condom, proper condom utilization may be affected by the state of mind during sexual intercourse. It is counter-intuitive to consider that improper use of a male condom during usage could be affected by the FSWs alcohol consumption level. However, during sexual intercourse with FSWs, it is not uncommon for FSWs to put the condom on the client.

**Table 5. Bivariate and multivariate logistic regression analysis showing the effect of monthly Heavy Episodic Drinking (HED) on outcome variables among female sex workers (FSW) in 11 towns in Ethiopia (2014).**

| Predictor variable | Dependent variables Crude OR (95% CI) | | | |
|---|---|---|---|---|
| | Physical beating | Condom breakage/slippage | Inconsistent condom use | HIV status |
| **HED** | | | | |
| No (ref) | 1.0 | 1.0 | 1.0 | 1.0 |
| Yes | 1.29 (1.11, 1.50) | 2.31(2.02, 2.64)) | 1.68 (1.35, 2.09) | 1.04 (0.90, 1.22) |
| **P-value** | 0.001 | < 0.001 | < 0.001 | 0.578 |
| | Adjusted OR (95% CI) | | | |
| | Physical beating | Condom breakage/slippage | Inconsistent condom use | |
| **HED** | | | | |
| No (ref) | 1.0 | 1.0 | 1.0 | |
| Yes | 1.27 (1.05, 1.53) | 1.44 (1.21, 1.70) | 1.30 (0.99, 1.72) | |
| **P-Value** | 0.014 | < 0.001 | 0.059 | |

Abbreviations: OR, odds ratios; CI, confidence interval.

HED was adjusted for sex selling starting age, average income, sex selling venue, khat chewing, start selling sex by force, and educational level variables

In our study, HIV infection was not significantly associated with HED, although the prevalence of HIV was high (20.8%). This finding is similar to results reported in studies conducted in Kenya [6, 10] but differs from other studies conducted in Botswana, South Africa, Rwanda, and Zambia which reported a significant association of HIV with HED [20–22]. Because of the cross-sectional study design, changes in alcohol consumption patterns over time could not be measured; therefore, we could not determine whether current drinking patterns differed from those at the time of HIV infection [6].

Another factor found to be significantly associated with HED was violence. Alcohol use is one of the most important risk indicators of increased violence against FSWs [42, 43].Alcohol use impairs FSWs' ability to detect the risk of violence and increases their vulnerability to risk-prone situations [27, 28].

Overall, our study shows that the proportion and associated consequences of HED among FSWs might require prevention and harm reduction efforts. Nevertheless, Ethiopia does not have any programs focusing on HED harm reduction among FSWs. Combining HED reduction programs with other ongoing programs such as HIV programs could help reduce HED and sustain epidemic control.

The strengths of this study included the large sample of FSWs from 11 cities and the sampling technique (RSD), which is recommended for hard-to-reach populations.

Our study had several limitations; the first is the findings are subject to social desirability bias. Secondly, we defined HED as ≥6 drinks per occasion, which might not be the definition used by other studies; this potential difference in definitions may have implications with regard to the risk factors that we identified. In addition, caution is needed when making inferences regarding causality; because this was a cross-sectional study, cause could not be

**Table 6. Heavy Episodic Drinking (HED) attributable risk fraction for the occurrence of physical beating, rape, condom breakage/slippage, and inconsistent condom use among female sex workers in 11 towns in Ethiopia (2014).**

| Predictor variable | Outcome variable | Attributable risk fraction (AF) (%) | Population attributable risk fraction (PAF) (%) |
|---|---|---|---|
| HED | Physical beating | 21.3 | 6.2 |
| | Condom breakage/slippage | 30.6 | 8.9 |

established. In addition, due to RDS method **FSWs with** small **network size might not be adequately represented**.

## Conclusion

We found that FSWs who started sex work before the age of 18 years, who had higher income from selling sex, who worked at venues where alcohol is sold, who were forced into sex work, and who chewed khat more frequently had increased likelihood of HED. In turn, HED increased the odds of violence, condom breakage, and inconsistent condom use. Increasing awareness of these factors and their consequences could help minimize the risks associated with HED for FSWs. In general, these results are likely to be relevant for FSW programs in other countries with a similar setting as Ethiopia and may inform targeted prevention strategies for FSWs.

## Supporting information

**S1 Questionnaire.**
(DOCX)

## Author Contributions

**Conceptualization:** Minilik Demissie Amogne, Anette Agardh, Ebba Abate, Benedict Oppong Asamoah.

**Formal analysis:** Minilik Demissie Amogne, Anette Agardh, Jelaludin Ahmed, Benedict Oppong Asamoah.

**Methodology:** Minilik Demissie Amogne, Jelaludin Ahmed.

**Supervision:** Minilik Demissie Amogne.

**Writing – original draft:** Minilik Demissie Amogne, Anette Agardh, Ebba Abate, Jelaludin Ahmed, Benedict Oppong Asamoah.

**Writing – review & editing:** Minilik Demissie Amogne, Anette Agardh, Ebba Abate, Jelaludin Ahmed, Benedict Oppong Asamoah.

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
