## [Decision Letter · Decision Letter 0]

9 Apr 2021

PONE-D-20-33642

Determinants and consequences of heavy episodic drinking among female sex workers in Ethiopia: A respondent-driven sampling study

PLOS ONE

Dear Dr. Amogne,

Thank you for submitting your manuscript to PLOS ONE. After careful consideration, we feel that it has merit but does not fully meet PLOS ONE’s publication criteria as it currently stands. Therefore, we invite you to submit a revised version of the manuscript that addresses the points raised during the review process.

Please note that I, as academic editor reviewed this paper as Reviewer 2. We did encounter challenges in getting a second reviewer which delayed the paper's disposition.  I agree with Reviewer 1 comments and suggestions.  

We look forward to receiving your revised manuscript.

Kind regards,

Kimberly Page, PhD, MPH

Academic Editor

PLOS ONE

Additional Editor Comments:

Thank you for your patience with the disposition of the article. Please note that the Academic Editor is Reviewer #2.

Journal Requirements:

3. Please include additional information regarding the survey or questionnaire used in the study and ensure that you have provided sufficient details that others could replicate the analyses. For instance, if you developed a questionnaire as part of this study and it is not under a copyright more restrictive than CC-BY, please include a copy, in both the original language and English, as Supporting Information.  If the original language is written in non-Latin characters, for example Amharic, Chinese, or Korean, please use a file format that ensures these characters are visible.

4. Please state whether you validated the questionnaire prior to testing on study participants. Please provide details regarding the validation group within the methods section.

"This research has been supported by the president’s plan for AIDS Relief (PEPFAR) through

Ethiopian Public Health Association (EPHA) under the terms of PS001229. The findings and

conclusions in this report are those of the authors and do not necessarily represent the official

position of the funding agencies."

6. We note that you have indicated that data from this study are available upon request. PLOS only allows data to be available upon request if there are legal or ethical restrictions on sharing data publicly. For information on unacceptable data access restrictions, please see http://journals.plos.org/plosone/s/data-availability#loc-unacceptable-data-access-restrictions.

Reviewers' comments:

Reviewer's Responses to Questions

**Comments to the Author**

1. Is the manuscript technically sound, and do the data support the conclusions?

Reviewer #1: Yes

Reviewer #2: Yes

2. Has the statistical analysis been performed appropriately and rigorously? 

Reviewer #1: Yes

Reviewer #2: Yes

3. Have the authors made all data underlying the findings in their manuscript fully available?

Reviewer #1: Yes

Reviewer #2: Yes

4. Is the manuscript presented in an intelligible fashion and written in standard English?

Reviewer #1: Yes

Reviewer #2: Yes

5. Review Comments to the Author

Reviewer #1: This is a solid study with a impressive sample size!

Some minor comments:

1. Background: Please explain the relationship of alcohol use to khat using some supporting evidence. It becomes more clear later (in the discussion when you mention that it masks the effects of intoxication) but it would be useful to understand this earlier in the paper.

2. You say, "although few studies have investigated the potential role of socio-demographic and

90 other contextual factors." But do not cite any. Please cite existing studies or reports from Ethiopia.

3. Limitations: You should mentions some internal validity issues associated with RDS.

4. Can you also mention some of the things that make the FSW environment in Ethiopia distinct from the other settings you compare to in the discussion? This may be helpful in the background as well.

5. Discussion: You recommend different types of targeted programming. Can you point to any HED reduction programs for FSW from any other settings?

Reviewer #2: Please note that the Academic Editor was Reviewer #2

This is a nicely presented paper which assessed heavy episodic drinking (HED) in a large sample of female sex workers (FSW) in Ethiopia. The research has many strengths including: systematic sampling method (RDS), the large diverse sample from multiple area of Ethiopia. This is a population highly impacted by HIV and the paper addresses an important exposure factor - alcohol use - which is associated with multiple individual and social correlates of HIV infection. The paper is well written. The background and methods are well presented. The results and tables are in great shape. I have a few minor comments that I hope will help the authors strengthen the paper.

1. Please consider replacing the word 'determinants' with 'correlates' throughout the paper when discussing "determinants of HED". Since this is a cross-sectional study, I think terminology should be changed. With respect to consequences - these are also correlates, but the authors might couch the language to indicate "potential consequences."

2. Intro, line 69: please add in some examples of the adverse health and socioeconomic consequences of Khat use. I also suggest saying earlier in the paragraph that this is a stimulant.

3. Methods, line 139: How were the independent variables chose/selected? From the literature? hypotheses? other?

4. Methods, line 145: similarly, how were these dependent variables or what was the rationale for selecting these?

5. Methods, line 179: did the authors assess whether missing data was random? Excluding this data if its is not random could lead to bias.

6. Results, line 218 here and elsewhere please change to correlates of HED. line 223: I suggest changing determinants to "independent correlates' of HED. I know some people use the word "on" when talking about models but I do prefer 'in'. (as in 'in analyses'. )

7. Discussion, lines 264-266: elaborate on this: was this a hypothesis?, or did the other authors show that alcohol use faciliates FSW's solicitation? It is also possible that women working in those high alcohol venues make more money. This potential confounding should be examined.

8. Discussion line 269: I think ou need to examine the colllinearity of venue and income - this could explain (be confounding) the association between income and HED. Research in Cambodia showed that women in bar settings drank more because they got commissions on how much alcohol they helped get sold. (And relative to the comment below about bar managers - line 274, they may be more concerned about alcohol income than women's health. )

9. line 274: I am concerned that this is a very big leap of a suggestion. I don't see anything in the data that makes this suggesting seem timely or feasible. I agree that programs are needed, but jumping to bar managers is another study. As noted above - in some settings the managers are invested in the women drinking more to sell more.

10. Line 300: please reframe high condom breakage/spillage: high relative to what?

11. Line 306: Unless you know for sure that FSW put condoms on clients consider - saying "it is not uncommon" or some such with less authority (this does sound anecdotal).

12. Line 310 - lack of association with HIV is also similar to findings from Cambodia.

13. Line 313-314 - very good point!

14: suggestion only: See Evans et al, Int J STD and AIDS 2021 about joint effects of alcohol and stimulants in FSW. And also, how other substances are associated with violence experienced by FSW (Draughon et al, DAD 2016) .

15. Table 2: please clarify the variable about "provide any regular financial support' [is this referring to family, or dependents, or?

6. PLOS authors have the option to publish the peer review history of their article (what does this mean?). If published, this will include your full peer review and any attached files.

Reviewer #1: **Yes: **Carinne Brody

Reviewer #2: No

---

## [Author Response · Author response to Decision Letter 0]

8 May 2021

Authors’ response to reviewer’s comments

Title: Determinants and consequences of heavy episodic drinking among female sex workers in Ethiopia: A respondent-driven sampling study in 11 major towns

Dear Editor and Reviewers,

Thank you for your comment and suggestions. We have revised the manuscript taking into account your comments. We have attached the revised version, and the responses point by point, are indicated below.

General comments 

Response: Thank you, we have now edited the manuscript based on the journal requirments

Response: Thank you, we included the statement below under the ethical consideration section

“Permission was obtained from the ethical review committee to collect consent from FSWs between the ages of 15 to 18 because they are considered to be emancipated minors”. Line 201

3. Please include additional information regarding the survey or questionnaire used in the study and ensure that you have provided sufficient details that others could replicate the analyses. For instance, if you developed a questionnaire as part of this study and it is not under a copyright more restrictive than CC-BY, please include a copy, in both the original language and English, as Supporting Information. If the original language is written in non-Latin characters, for example Amharic, Chinese, or Korean, please use a file format that ensures these characters are visible.

Response: Thank you, we included the English version of the questionnaire as a supporting document since it is not restricted under any copyright. The Amharic version is no longer available

4. Please state whether you validated the questionnaire prior to testing on study participants. Please provide details regarding the validation group within the methods section.

Response: Thank you, we included a statement mentioning the pilot process under the ‘data collection procedure’ section

“The questionnaire was piloted before the actual implementation of the study in a town that was not included as a site” Line 125

"This research has been supported by the president’s plan for AIDS Relief (PEPFAR) through

Ethiopian Public Health Association (EPHA) under the terms of PS001229. The findings and

conclusions in this report are those of the authors and do not necessarily represent the official

position of the funding agencies."

Response: Thank you, we excluded all funding statement from the manuscript. For the current publication process, they provide no funding but the study was funded by PEPFAR. The data set was obtained from the Ethiopian Public Health Institute (EPHI). Thus please edit the funding statement as follows:_

“The study was funded by the president’s plan for AIDS Relief (PEPFAR) through the Ethiopian Public Health Association (EPHA) under the terms of PS001229”. We will mention this in the revised cover letter.

6. We note that you have indicated that data from this study are available upon request. PLOS only allows data to be available upon request if there are legal or ethical restrictions on sharing data publicly. For information on unacceptable data access restrictions, please see http://journals.plos.org/plosone/s/data-availability#loc-unacceptable-data-access-restrictions.In your revised cover letter, please address the following prompts:

Response: Thank you. the data are owned by the Ethiopian public health institute (EPHI) and therefore there is restricted public access. Anyone who wants to obtain data from the institute should write an email to the director general copying the director of data managment (Dr. Alemnesh). Their details are below, and we have updated this in the cover letter.

Director general email: Dr. Ebba: ebbaabate@yahoo.com

Dr. Alemnesh: alemmirkuzie@yahoo.com

Reviewer #1: Comment and response

1. Background: Please explain the relationship of alcohol use to khat using some supporting evidence. It becomes more clear later (in the discussion when you mention that it masks the effects of intoxication) but it would be useful to understand this earlier in the paper.

Response: Thank you, we revised the statement to clarify the releationship.

‘Young people often chew khat during the daytime and go to bars to drink alcohol at night to purposely reduce the stimulant effect” Line 69

2. You say, "although few studies have investigated the potential role of socio-demographic and

 other contextual factors." But do not cite any. Please cite existing studies or reports from Ethiopia.

Response: Thank you, we inserted citations as per your comment. Line 92

3. Limitations: You should mentions some internal validity issues associated with RDS.

Response: Thank you, we included a limitation of RDS as mentioned below.

“In addition, due to the RDS method, FSWs with small network size might not be adequately represented”. Line 354

4. Discussion: You recommend different types of targeted programming. Can you point to any HED reduction programs for FSW from any other settings?

Response: Although we couldn’t identify indicate a specific program directed towards FSWS in Ethiopia,, intervention studies conducted in different countries such as Mexico and Kenya suggested that alcohol reduction could contribute to violence reduction and recommended combining with HIV councelling to achieve good results. 

Reviewer #2: Comment and response

This is a nicely presented paper which assessed heavy episodic drinking (HED) in a large sample of female sex workers (FSW) in Ethiopia. The research has many strengths including: systematic sampling method (RDS), the large diverse sample from multiple area of Ethiopia. This is a population highly impacted by HIV and the paper addresses an important exposure factor - alcohol use - which is associated with multiple individual and social correlates of HIV infection. The paper is well written. The background and methods are well presented. The results and tables are in great shape. I have a few minor comments that I hope will help the authors strengthen the paper.

1. Please consider replacing the word 'determinants' with 'correlates' throughout the paper when discussing "determinants of HED". Since this is a cross-sectional study, I think terminology should be changed. With respect to consequences - these are also correlates, but the authors might couch the language to indicate "potential consequences."

Response: Thank you, we agree that both the determinants and consequences are correlates of HED as far as this is a cross-sectional study. However, we felt that it was meaningful to regard certain background factors that had an association with HED as likely determinants, due to the underlying nature of these measures and also to findings from previous studies cited in the manuscript. For the same reasons, we regarded other measures as very likely to be potential consequences of HED rather than determinants, and accordingly, we examined them separately. Therefore, we prefer to use determinant and consequences to separate the two sets of analyses; otherwise, both are “correlates” of HED. 

2. Intro, line 69: please add in some examples of the adverse health and socioeconomic consequences of Khat use. I also suggest saying earlier in the paragraph that this is a stimulant.

Response: Thank you, we have inserted a statement on the health and economic effects of khat chewing. under line 65-70

3. Methods, line 139: How were the independent variables chose/selected? From the literature?hypotheses? other?

Response: Thank you, we inserted a statement describing how the independent variables were selected.

4. Methods, line 145: similarly, how were these dependent variables or what was the rationale for selecting these?

Response: Thank you, we inserted a statement describing how the dependent variables were selected. Line 147

5. Methods, line 179: did the authors assess whether missing data was random? Excluding this data if its is not random could lead to bias.

Response: Thank you, we assessed this during the analysis, and the missing data was randomly distributed. Line 147

6. Results, line 218 here and elsewhere please change to correlates of HED. line 223: I suggest changing determinants to "independent correlates' of HED. I know some people use the word "on" when talking about models but I do prefer 'in'. (as in 'in analyses'. )

Response: Thank you, we prefer to use the terms predictors/determinants and consequences, as explained above in response #1. We have changed the word “on” to “in”, as suggested. Line 226, 236

7. Discussion, lines 264-266: elaborate on this: was this a hypothesis?, or did the other authors show that alcohol use faciliates FSW's solicitation? It is also possible that women working in those high alcohol venues make more money. This potential confounding should be examined.

Response: Thank you, the study was a mixed study conducted in Malawi and reported tha alcohol sale facilitates the negotiation with clients, and the reference is provided. Other studies also suggested that clients prefer FSWs who drink alcohol, so FSWs tend to drink alcohol. We feel that the text concerning this is satisfactory and does not require further explanation.

8. Discussion line 269: I think you need to examine the colllinearity of venue and income - this could explain (be confounding) the association between income and HED. Research in Cambodia showed that women in bar settings drank more because they got commissions on how much alcohol they helped get sold. (And relative to the comment below about bar managers - line 274, they may be more concerned about alcohol income than women's health. )

Response: Thank you, as mentioned in the “data analysis” section, we conducted correlation analysis to assess for multicollinearity and all variables included in the analysis did not have any multicollinearity. Regarding commission from the drinking, as far as my knowledge, in the bars in Ethiopia there is no such a thing. They just use the bars to meet clients and pay for the bar bill when they leave with clients.

9. line 274: I am concerned that this is a very big leap of a suggestion. I don't see anything in the data that makes this suggesting seem timely or feasible. I agree that programs are needed, but jumping to bar managers is another study. As noted above - in some settings the managers are invested in the women drinking more to sell more.

Response: Thank you, we agree with the suggestion, and we revised the sentence.

10. Line 300: please reframe high condom breakage/spillage: high relative to what?

Response: Thank you, the comparison was with the condom use reported,. Almost all reported consistent condom use but 25% experienced condom breakage/slippage in the past one month. Therefore, the sentence is intented to illustrate the problem represented by condom breakage, although they are using condoms.

11. Line 306: Unless you know for sure that FSW put condoms on clients consider - saying "it is not uncommon" or some such with less authority (this does sound anecdotal).

 Response: Thank you, we revised this accordingly. Line 331

12. Table 2: please clarify the variable about "provide any regular financial support' [is this referring to family, or dependents, or?

Response: Thank you, the support concerns anyone, including a family member, but the emphasis is on the fact that the support is regular. We included a word regarding this in the table

Note: On the process of addressing the comment and suggestions provided, additional references are included in the manuscript.

---

## [Editor Report · Decision Letter 1]

17 May 2021

Determinants and consequences of heavy episodic drinking among female sex workers in Ethiopia: A respondent-driven sampling study

PONE-D-20-33642R1

Dear Dr. Amogne,

We’re pleased to inform you that your manuscript has been judged scientifically suitable for publication and will be formally accepted for publication once it meets all outstanding technical requirements.

Kind regards,

Kimberly Page, PhD, MPH

Academic Editor

PLOS ONE
---

## [Editor Report · Acceptance letter]

20 May 2021

PONE-D-20-33642R1 

Determinants and consequences of heavy episodic drinking among female sex workers in Ethiopia: A respondent-driven sampling study 

Dear Dr. Amogne:

I'm pleased to inform you that your manuscript has been deemed suitable for publication in PLOS ONE. Congratulations! Your manuscript is now with our production department. 

Kind regards, 

on behalf of

Dr. Kimberly Page 

Academic Editor

PLOS ONE